# The epidemiology of HIV population viral load in twelve sub-Saharan African countries

**Wolfgang Hladik**[1]*, **Paul Stupp**[1], **Stephen D. McCracken**[1], **Jessica Justman**[2], **Clement Ndongmo**[1], **Judith Shang**[1], **Emily K. Dokubo**[1], **Elizabeth Gummerson**[2], **Isabelle Koui**[3], **Stephane Bodika**[1], **Roger Lobognon**[1], **Hermann Brou**[2], **Caroline Ryan**[1], **Kristin Brown**[1], **Harriet Nuwagaba-Biribonwoha**[2], **Leonard Kingwara**[4], **Peter Young**[1], **Megan Bronson**[1], **Duncan Chege**[2], **Optatus Malewo**[1], **Yohannes Mengistu**[1], **Frederix Koen**[2], **Andreas Jahn**[5], **Andrew Auld**[1], **Sasi Jonnalagadda**[1], **Elizabeth Radin**[2], **Ndapewa Hamunime**[6], **Daniel B. Williams**[1], **Eugenie Kayirangwa**[1], **Veronicah Mugisha**[2], **Rennatus Mdodo**[1], **Stephen Delgado**[2], **Wilford Kirungi**[7], **Lisa Nelson**[1], **Christine West**[1], **Samuel Biraro**[2], **Kumbutso Dzekedzeke**[2], **Danielle Barradas**[1], **Owen Mugurungi**[8], **Shirish Balachandra**[1], **Peter H. Kilmarx**[1], **Godfrey Musuka**[2], **Hetal Patel**[1], **Bharat Parekh**[1], **Katrina Sleeman**[1], **Robert A. Domaoal**[1], **George Rutherford**[9], **Tsietso Motsoane**[10], **Anne-Cécile Zoung-Kanyi Bissek**[11,12], **Mansoor Farahani**[2], **Andrew C. Voetsch**[1]

**1** Division of Global HIV and TB, US Centers for Disease Control and Prevention (CDC), Atlanta, GA, United States of America, **2** ICAP at Columbia University, New York, New York, United States of America, **3** Ministry of Health, Abidjan, Cote d'Ivoire, **4** National AIDS and STI's Control Programme, Ministry of Health, Nairobi, Kenya, **5** Ministry of Health, Lilongwe, Malawi, **6** Ministry of Health, Windhoek, Namibia, **7** Ministry of Health, Kampala, Uganda, **8** Ministry of Health and Child Care, Harare, Zimbabwe, **9** University of California San Francisco, San Francisco, California, United States of America, **10** Ministry of Health, Maseru, Lesotho, **11** Division of Operational Research for Health, Ministry of Health, Yaoundé, Cameroon, **12** Faculty of Medicine and Biomedical Sciences, University of Yaoundé, Yaoundé, Cameroon

* wfh3@cdc.gov

**Data Availability Statement:** The data underlying the results presented in the study are available from https://phia-data.icap.columbia.edu/datasets. Information about how to access and download

## Abstract

### Background

We examined the epidemiology and transmission potential of HIV population viral load (VL) in 12 sub-Saharan African countries.

### Methods

We analyzed data from Population-based HIV Impact Assessments (PHIAs), large national household-based surveys conducted between 2015 and 2019 in Cameroon, Cote d'Ivoire, Eswatini, Kenya, Lesotho, Malawi, Namibia, Rwanda, Tanzania, Uganda, Zambia, and Zimbabwe. Blood-based biomarkers included HIV serology, recency of HIV infection, and VL. We estimated the number of people living with HIV (PLHIV) with suppressed viral load (<1,000 HIV-1 RNA copies/mL) and with unsuppressed viral load (viremic), the prevalence of unsuppressed HIV (population viremia), sex-specific HIV transmission ratios (number female incident HIV-1 infections/number unsuppressed male PLHIV per 100 persons-years [PY] and vice versa) and examined correlations between a variety of VL metrics and incident HIV. Country sample sizes ranged from 10,016 (Eswatini) to 30,637 (Rwanda); estimates were weighted and restricted to participants 15 years and older.

PHIA data can be found here: https://phia-data.icap.columbia.edu/, including through this instructional video: https://phia-data.icap.columbia.edu/about

**Funding:** This project has been supported by the President's Emergency Plan for AIDS Relief (PEPFAR) through the Centers for Disease Control and Prevention under the terms of cooperative agreements #U2GGH001226 and U2GGH001271. The funders had no role in study design, data collection and analysis, decision to publish, or preparation of the manuscript.

**Competing interests:** The authors have declared that no competing interests exist.

## Results

The proportion of female PLHIV with viral suppression was higher than that among males in all countries, however, the number of unsuppressed females outnumbered that of unsuppressed males in all countries due to higher overall female HIV prevalence, with ratios ranging from 1.08 to 2.10 (median: 1.43). The spatial distribution of HIV seroprevalence, viremia prevalence, and number of unsuppressed adults often differed substantially within the same countries. The 1% and 5% of PLHIV with the highest VL on average accounted for 34% and 66%, respectively, of countries' total VL. HIV transmission ratios varied widely across countries and were higher for male-to-female (range: 2.3–28.3/100 PY) than for female-to-male transmission (range: 1.5–10.6/100 PY). In all countries mean $\log_{10}$ VL among unsuppressed males was higher than that among females. Correlations between VL measures and incident HIV varied, were weaker for VL metrics among females compared to males and were strongest for the number of unsuppressed PLHIV per 100 HIV-negative adults ($R^2 = 0.92$).

## Conclusions

Despite higher proportions of viral suppression, female unsuppressed PLHIV outnumbered males in all countries examined. Unsuppressed male PLHIV have consistently higher VL and a higher risk of transmitting HIV than females. Just 5% of PLHIV account for almost two-thirds of countries' total VL. Population-level VL metrics help monitor the epidemic and highlight key programmatic gaps in these African countries.

## Introduction

Globally, an estimated 38 million people were living (PLHIV) with HIV in 2021 [1]. Sub-Saharan Africa (SSA), and in particular Eastern and Southern Africa, is the epicenter where most HIV epidemics are generalized and severe [1]. The massive scale-up of antiretroviral therapy (ART) beginning in the early-to-mid-2000s in SSA led to a dramatic reduction in HIV-related mortality and a slower decline in adult HIV incidence [1]. In the absence of a vaccine or cure, Treatment as Prevention, a concept informed by clinical trials demonstrated that ART-mediated viral load suppression (VLS) minimizes the risk of onward sexual transmission of HIV [2]. Using modeled findings, UNAIDS suggests that by meeting first the global 90-90-90 targets by 2020 and by 2025 the global 95-95-95 targets (95% of all PLHIV are aware of their HIV-positive status, among PLHIV aware of their HIV positive status, 95% are on ART; and, among those on ART, 95% have VLS), will help "end AIDS" by 2030 [3]. VLS reduces the HIV-related mortality and sexual transmission of HIV. These expected dual population-level effects of ART on mortality and HIV incidence make monitoring viral load (VL) essential.

However, the utility of VL metrics go beyond monitoring progress toward the third 95 target; notably VL also lends itself to inform a population's (remaining) HIV transmission potential. Population viremia, the population prevalence of unsuppressed HIV often is strongly correlated with higher HIV incidence in the larger population [4–6]. Unfortunately, VL data that consider only PLHIV in the denominator as well as clinic-based VL metrics that exclude undiagnosed or untreated PLHIV often show statistically weak or insignificant relationships with HIV incidence in the background population [4, 7]. Further, while dichotomizing VL into suppressed and unsuppressed has utility for programming and policy, the categorization also leads to

information loss as the level of VL is strongly correlated with HIV disease progression and transmission probability. Numerous VL metrics are described in the literature, and efforts to use such metrics are increasing. Examples include the mean and total "community viral load" as "markers of access to care and treatment" and indicators of a population's viral burden, [6, 8, 9] or, borrowing a graphical concept from economics, the Lorenz curve (and associated Gini coefficient) help to display the often highly unequal distribution of VL in a given population as exemplified in a San Francisco patient cohort [10]. Above all, VL-related metrics allow describing the HIV transmission risk from PLHIV to uninfected individuals [11, 12].

As population-level VL data appear key to understanding progress towards effective HIV control, such data are becoming increasingly common in SSA countries, e.g., in Uganda and Kenya [4, 13, 14]. We examined data from 12 national household-based surveys in SSA with the goal to describe the epidemiology of VL and to examine the utility of various VL metrics for surveillance and epidemic monitoring at the national population level.

## Materials and methods

### Data sources, setting and survey design

PEPFAR, the President's Emergency Plan for AIDS Relief, a U.S. Government program assisting HIV response efforts in selected low- and middle-income countries, funded a series of national household-based surveys in 2015–2019 to assess the impact of HIV programs in SSA countries. Implementing partners included country governments, ICAP at Columbia University in New York, and the Centers for Disease Control and Prevention (CDC). The field methods of these Population-based HIV Impact Assessments (PHIAs) are described in detail elsewhere [15, 16]. Briefly, PHIAs were powered for national adult HIV incidence estimation as well as sub-national VLS. A two-stage cluster sampling design was used to select clusters and households. The PHIA data sets are in the public domain and available upon request [17].

### Study population

We analyzed PHIA data from Cameroon (2017–18, number of adult respondents with an unambiguous HIV result in the survey: 26,031), Cote d'Ivoire (2017–18, n = 17,813), Eswatini (2016–17, n = 10,016), Kenya (2018–19, n = 27,745), Lesotho (2016–17, n = 11,682), Malawi (2015–16, n = 17,187), Namibia (2017, n = 16,939), Rwanda (2018–19, n = 30,637), Tanzania (2016–17, n = 29,369), Uganda (2016–17, n = 29,024), Zambia (2016, n = 19,115), and Zimbabwe (2015–16, n = 20,577). The PHIAs sampled PLHIV irrespective of their reported serostatus knowledge, linkage to care, or retention in care. Among participants enrolled in the surveys the proportion who consented to a blood draw ranged from 87.5% to 99.7% (median: 92.5%). Our study considered male and female survey participants aged 15–64 years (except Lesotho and Zambia: 15–59 years) for whom an HIV test was conducted during the survey. Additional survey eligibility criteria included having spent the last night in the sampled household and speaking one of the offered interview languages. We excluded children from the analysis due to their much-diminished HIV transmission potential and because the number of sampled HIV-infected children was small in each country. We excluded the Ethiopia PHIA as sampling was restricted to urban areas and so did not allow for the examination of national VL metrics or the relationship between viremia and HIV incidence.

### Data collection

Interview data were collected through structured face-to-face interviews at the households using mobile tablets. Original data variables included household characteristics, demographics

(including their sex at birth, hence the terms *male* and *female*), HIV-related service uptake, and HIV-related risk behaviors. Participants were asked about their HIV testing history, past test results, and ART status.

## Biologic measures

Detailed descriptions of our biomarker procedures have been published elsewhere [18]. Venous blood collected with Ethylenediaminetetraacetic acid (EDTA) as an anticoagulant was tested at the household for HIV using the countries' national rapid test algorithm. All HIV-positive specimens underwent confirmatory testing using the Geenius HIV 1/2 Supplemental Assay (Bio-Rad, Hercules, CA, USA) or PCR testing. Additional HIV quality control testing varied by country. To determine VL, HIV-positive plasma specimens (or, if unavailable, dried blood spots (DBS), accounting for 3% of all participants) were further tested in-country using a Roche or Abbott platform [19]. The lower limit of detection was 20–40 copies/mL for plasma and 700–800 copies/mL for DBS. HIV-positive participants with a VL<1,000 HIV-1 RNA copies/mL were classified as virally suppressed; those with a VL≥1,000 copies/mL were classified as unsuppressed or viremic.

HIV-positive participants (except in Kenya and Rwanda) received a CD4+ T-cell count measurement in the field using the Pima™ CD4 Analyzer (Abbott Molecular Inc., Chicago, IL, USA, formerly Alere).

DBS from HIV-positive participants were qualitatively tested for the presence of three antiretroviral (ARV) drugs (two from the countries' prevailing 1st line ARV regimens, plus one from the prevailing 2nd line regimens) using high performance liquid chromatography coupled with tandem mass spectrometry at the University of Cape Town, South Africa.

For HIV-1 recency testing, HIV-seropositive specimens were tested in-country with the limiting antigen (LAg) avidity enzyme immunoassay [20]. Specimens with a normalized optical density (ODn) value of ≤2.0 were re-tested for HIV recency in triplicate; median ODn values of ≤1.5 were classified as LAg-recent. For DBS specimens, the Maxim HIV-1 LAg DBS EIA was used; median values of ≤1.0 were classified as LAg-recent. Participants who tested LAg-recent, had a VL≥1,000 copies/mL, and tested ARV-negative were classified as having recent HIV-1-infection. Participants with a LAg ODn of >1.5 or a VL<1,000 copies/mL, or had detectable ARV were classified as having long-term HIV-1 infection. More detail is published elsewhere [21].

## Data management and analysis

The data presented for each survey are based on participants with a valid biomarker result for HIV-1 status (HIV-2 status was not considered for the definition for HIV status and prevalence). The analytic weights account for the selection probabilities of primary sampling units, households, and individuals and incorporate adjustments for non-response by households, individuals, and biomarker participation. The weights were further adjusted to match each country's age and sex distribution from a recent population projection. Estimates across all countries combined were also weighted by population size. We classified HIV-infected participants as diagnosed or aware of their HIV status if they reported a prior HIV positive test result or if they had detectable ARV. HIV-positive participants who tested ARV-positive were classified as being on ART.

For each country, we estimated HIV incidence based on the number of HIV-recent test results, mean duration of recency (130 days for most countries), time cutoff of one year, and a proportion false recent of 0.00 for the LAg avidity assay. Other computed indicators included the arithmetic mean VL, the number of suppressed and unsuppressed PLHIV by sex and age,

countries' total VL (weighted national sum-total of all PLHIV's VL in a country, using 1 c/mL for undetectable VL). We computed VLS for all PLHIV (population VLS), diagnosed PLHIV, and PLHIV on ART. We computed the proportion virally suppressed among PLHIV who tested ARV-negative as well as the proportion unsuppressed among PLHIV who tested ARV-positive. We estimated population viremia by multiplying HIV prevalence by the proportion PLHIV who were unsuppressed (viremic).

We mapped HIV and viremia prevalence using geomasked enumeration area centroids [22] and spatial interpolation methods to explore the spatial dimensions of viremia. Specifically, we used kernel density smoothing within PrevR [23, 24]. A surface was computed using a Gaussian kernel which depends on each enumeration area's number of HIV (or unsuppressed) cases and all tested cases. The "intensity" surfaces (number of persons per area) were calculated separately for each (weighted number of HIV-infected, unsuppressed [numerators] and tested individuals [denominator]) and a ratio was calculated from these to produce smooth surfaces of viremia and HIV prevalence. The results of this modeling were reworked in QGIS 3.4 [25] and ArcMap 10.5.1 (Environmental Systems Research Institute, Redlands, CA, USA) and presented with country-specific scales to each country's minimum and maximum values using percent stretch with 0.5% clipping; this country-specific strategy highlights local differences in spatial pattern in HIV and viremia prevalence [26, 27]. To visualize the spatial pattern of PLHIV and unsuppressed PLHIV we applied raster math of each prevalence surface and the 2020 spatial population models (unconstrained, UN adjusted, 100m resolution) for each country from the WorldPop [28] program. Minor adjustments were made to rescale the resultant product totals to PHIA weighted estimates of adult population, PLHIV, and number of unsuppressed persons at the national level (15–59 or 15–64 years [see Table 1]). Adjustments are presented in S1 Table. The maps displaying the number of unsuppressed persons per pixel are produced using a "histogram equalize" stretch to increase contrast for visualization.

We constructed Lorenz curves [29] and computed Gini coefficients [30] for VL by country using SAS v9.4 (SAS Institute, Cary, NC, USA) to construct population-weighted cumulative percentiles of total VL that correspond to percentiles of the population. In ART-naïve HIV-infected populations one would expect a scattered distribution of mostly unsuppressed VL values, however, in the presence of widespread treatment one can expect a growing disparity ranging from undetectable to high VL values. Undetectable VL values were assigned a VL value of 1 for the construction of the Lorenz curves. Gini coefficients are a measure of statistical dispersion, ranging from 0 (perfect parity) to 1 (perfect disparity), whereas Lorenz curves graphically depict disparity. For the computation of Gini coefficients and construction of Lorenz curves, we included all HIV-infected participants and ordered the observations by the weighted number of VL copies.

For each country we estimated the annualized HIV transmission ratios using the following formula: (estimated number of annual new HIV-1 infections/number PLHIV)* 100, thus deriving estimates of the number of HIV transmissions per 100 PLHIV per year. Assuming that all transmissions stem from unsuppressed PLHIV we further estimated *effective* HIV transmission ratios using the same formula but restricting the number of PLHIV to unsuppressed only. Assuming that all transmission was between opposite sexes we computed sex-specific effective HIV transmission ratios (male-to-female and female-to-male). Using the same data, we also estimated the number of unsuppressed adults per incident HIV infection for each country by dividing the number of unsuppressed adults by the number of incident HIV infections, for both sexes combined as well as for each sex (No. unsuppressed males / No. female incident HIV; No. unsuppressed females / No. male incident HIV), thus deriving a

**Table 1. HIV prevalence, incidence, and viral load metrics by country.** Population-based HIV Impact Assessments, 2015–19.

| | Cameroon | Cote d'Ivoire | Eswatini | Kenya | Lesotho | Malawi | Namibia | Rwanda | Tanzania | Uganda | Zambia | Zimbabwe | Median |
|---|---|---|---|---|---|---|---|---|---|---|---|---|---|
| Age range (years) | 15–64 | 15–64 | 15–64 | 15–64 | 15–59 | 15–64 | 15–64 | 15–64 | 15–64 | 15–64 | 15–59 | 15–64 | N/A |
| HIV prevalence (%) | 3.7 | 2.8 | 27.9 | 4.9 | 25.6 | 10.6 | 12.6 | 3.0 | 5.0 | 6.3 | 12.0 | 14.1 | 8.4 |
| No. PLHIV, total[1] | 499,863 | 381,907 | 192,387 | 1,303,267 | 305,853 | 901,341 | 176,329 | 210,200 | 1,494,555 | 1,195,300 | 960,665 | 1,152,520 | 700,602 |
| No. PLHIV suppressed | 223,308 | 154,505 | 140,138 | 932,846 | 206,874 | 617,071 | 136,472 | 159,784 | 778,066 | 712,653 | 577,628 | 687,728 | 400,468 |
| No. PLHIV unsuppressed | 276,556 | 227,401 | 52,249 | 370,421 | 98,979 | 284,271 | 39,857 | 50,416 | 716,490 | 482,647 | 383,037 | 464,792 | 280,413 |
| Ratio female: male unsuppressed PLHIV | 2.10 | 1.64 | 1.43 | 1.58 | 1.17 | 1.08 | 1.10 | 1.28 | 1.42 | 1.45 | 1.54 | 1.15 | 1.43 |
| Proportion female among unsuppressed PLHIV | 67.7% | 62.1% | 58.8% | 61.2% | 53.9% | 51.9% | 52.4% | 56.1% | 58.7% | 59.2% | 60.6% | 53.5% | 58.8% |
| Mean VL ($log_{10}$), all PLHIV | 2.86 | 3.15 | 1.56 | 1.67 | 1.95 | 1.55 | 1.51 | 1.40 | 2.66 | 2.29 | 2.16 | 2.08 | 2.02 |
| Mean VL ($log_{10}$), unsuppressed PLHIV | | | | | | | | | | | | | |
| Both sexes | 4.88 | 4.73 | 4.51 | 4.49 | 4.49 | 4.36 | 4.67 | 4.42 | 4.63 | 4.62 | 4.70 | 4.60 | 4.61 |
| Male | 5.15 | 4.75 | 4.60 | 4.52 | 4.54 | 4.48 | 4.77 | 4.47 | 4.85 | 4.78 | 4.80 | 4.71 | 4.73 |
| Female | 4.75 | 4.72 | 4.45 | 4.47 | 4.44 | 4.25 | 4.59 | 4.39 | 4.48 | 4.51 | 4.63 | 4.50 | 4.49 |
| Viral load suppression (VLS, %) | | | | | | | | | | | | | |
| Both sexes | 44.7 | 40.2 | 72.8 | 71.6 | 67.6 | 68.3 | 77.4 | 76.0 | 52.0 | 59.6 | 59.2 | 59.6 | 63.6 |
| Male | 42.5 | 27.7 | 66.9 | 65.1 | 63.4 | 60.9 | 69.6 | 70.5 | 41.2 | 53.6 | 57.2 | 53.6 | 59.1 |
| Female | 45.6 | 45.9 | 75.9 | 74.6 | 70.5 | 73.1 | 81.7 | 79.1 | 57.5 | 62.9 | 60.4 | 63.7 | 67.1 |
| Diagnosed VLS | 74.5 | 68.1 | 82.2 | 87.0 | 81.2 | 83.8 | 88.6 | 88.2 | 78.8 | 76.3 | 79.1 | 76.1 | 80.2 |
| In-treatment VLS | 80.1 | 73.7 | 91.3 | 90.6 | 87.7 | 91.3 | 91.3 | 90.1 | 87.0 | 83.7 | 89.2 | 85.3 | 88.5 |
| Prevalence of detectable viremia | 2.0 | 1.7 | 7.6 | 1.4 | 8.3 | 3.3 | 2.8 | 0.7 | 2.4 | 2.5 | 4.8 | 5.7 | 2.7 |
| Among ARV-positives, proportion unsuppressed (%) | 17.1 | 21.1 | 6.1 | 6.3 | 9.1 | 6.2 | 6.6 | 7.2 | 11.0 | 11.6 | 8.1 | 12.1 | 8.6 |
| Among ARV-negatives, proportion suppressed (%) | 9.6 | 12.1 | 13.4 | 21.2 | 10.7 | 14.2 | 20.7 | 17.1 | 8.9 | 16.1 | 9.2 | 9.9 | 12.8 |
| Total VL (billion copies/mL) | 78.6 | 40.6 | 5.7 | 48.9 | 9.3 | 23.5 | 8.3 | 5.3 | 101.8 | 80.8 | 70.0 | 60.6 | 44.7 |
| Total VL ($log_{10}$) | 10.90 | 10.61 | 9.76 | 10.69 | 9.97 | 10.37 | 9.92 | 9.73 | 11.01 | 10.91 | 10.85 | 10.78 | 10.65 |
| Percent of total VL due to highest 1% of PLHIV | 22.2 | 17.1 | 42.4 | 42.8 | 33.3 | 33.7 | 46.5 | 47.2 | 23.7 | 41.4 | 34.7 | 34.0 | 34.3 |
| Percent of total VL due to highest 5% of PLHIV | 49.0 | 45.9 | 73.7 | 71.6 | 66.0 | 68.8 | 77.6 | 77.0 | 56.1 | 67.0 | 63.2 | 63.6 | 66.5 |
| Gini coefficient (PLHIV only) | 0.816 | 0.788 | 0.918 | 0.918 | 0.893 | 0.900 | 0.932 | 0.931 | 0.836 | 0.890 | 0.881 | 0.875 | 0.892 |
| Gini coefficient (all adults) | 0.994 | 0.994 | 0.979 | 0.996 | 0.975 | 0.990 | 0.992 | 0.998 | 0.992 | 0.994 | 0.987 | 0.983 | 0.992 |

HIV: human immunodeficiency virus; $log_{10}$: log base 10; PLHIV: people living with HIV (Note: No. PLHIV are computed, weighted estimates); PY: person-years; VLS: viral load suppression; ARV: antiretroviral (drugs). The denominator for the prevalence of detectable viremia includes both HIV-infected and uninfected adults. "Highest 1%" and "Highest 5%" refer to the 1% and 5% of PLHIV with the highest viral load.

measure of how many unsuppressed adults/males/females on average lead to one incident case of adult/female/male HIV per year.

We examined the relationship between incident HIV and various metrics across the 12 countries by computing the coefficient of determination ($R^2$). The examined metrics included total VL (computed as a country's estimated sum of all PLHIV's VL, absolute and $log_{10}$ transformed), population viremia, mean VL (per capita VL, i.e., HIV-infected, and uninfected adults combined), Gini coefficients, VLS (for all PLHIV, for diagnosed PLHIV, and for PLHIV on ART) and, for comparison purposes, HIV seroprevalence. For all metrics we computed $R^2$ values for both sexes combined; for select metrics we also computed $R^2$ values for opposite sex correlations to examine male-to-female and female-to-male transmission potentials. For this sub-analysis we re-computed Gini coefficients by including all participants, both HIV-

uninfected and infected, assigning HIV-uninfected adults a $\log_{10}$ value of 0 (i.e., an absolute VL value of 1 copy/mL).

### Ethics statement

The survey protocols were approved by appropriate institutional review boards in each country, at Columbia University, Westat, UCSF (for Namibia), and at CDC. Institutional review board names, protocol numbers, and approval dates for each survey are shown as Supporting Information. Written informed consent was obtained from adult participants; assent and parental permission was obtained for participants under the age of consent (typically 15–17 years). HIV tests were returned to participants (In Kenya, participants could also opt to receive their HIV results at a health facility of their choice.); VL results were communicated to the health facility chosen by the participants. Efforts were made to link HIV-infected participants not on ART to a health care provider of their choice.

## Results

### Descriptive viral load metrics

Table 1 shows selected HIV and VL-related metrics; corresponding measures of uncertainty are presented in S2 Table. The estimated population viremia (population prevalence of unsuppressed HIV) ranged from 0.7% (Rwanda) to 8.3% (Lesotho). Tanzania was estimated to have the largest total VL (101.8 billion copies/mL), whereas Rwanda's total VL offered the smallest estimate at 5.3 billion copies/mL. $\log_{10}$ transformed mean VL among unsuppressed PLHIV ranged from 4.36 (Malawi) to 4.88 (Cameroon), corresponding to an absolute VL range of 22,900 to 75,900 copies/mL. In each country examined, the mean $\log_{10}$ VL among unsuppressed males (range across countries: 4.47–5.15) exceeded that among unsuppressed females (range across countries: 4.25–4.75); however, 95% CI for some countries overlapped (S2 Table). The highest mean $\log_{10}$ VL for both males and females were observed in Cameroon.

Fig 1 displays the frequency distribution of categorical VL among ARV-negative PLHIV by sex, combined for all 12 countries. Females more frequently showed a VL category below 50,000 copies/mL (peak frequency 1,000<10,000 c/ml) whereas males more often showed a VL category above 50,000 copies/mL (peak frequency 100,000<250,000 c/ml). Fig 2 displays the $\log_{10}$ VL by CD4+ T cell count category and sex; male ARV-negative PLHIV tended to exhibit higher VL than females. Fig 3 shows the country-specific population pyramids for PLHIV by age band and viral suppression status, illustrating substantial variation in the distribution of viremia by sex and age. In most countries the estimated number of both male and female unsuppressed adults increased from age 15–19 years onwards, often peaking around age 30–35 years and declining again thereafter. In all countries, female unsuppressed PLHIV outnumbered male unsuppressed PLHIV (Table 1, median: 59%). The ratio of female:male unsuppressed adults varied from 1.08 (Malawi) to 2.10 (Cameroon).

Fig 4 displays the spatial distribution of estimated HIV seroprevalence, viremia prevalence, and the number of unsuppressed PLHIV. Differences across maps should be appreciated within the same country (rather than across countries) and are a function of the proportion of PLHIV suppressed (affecting viremia prevalence) as well as population density (also affecting the number unsuppressed PLHIV).

### Lorenz curves and Gini coefficients

Fig 5 shows the Lorenz curves (cumulative No. of PLHIV vs. ordered cumulative total VL) for each country. The curves' shapes across the 12 countries were similar, reflecting the large

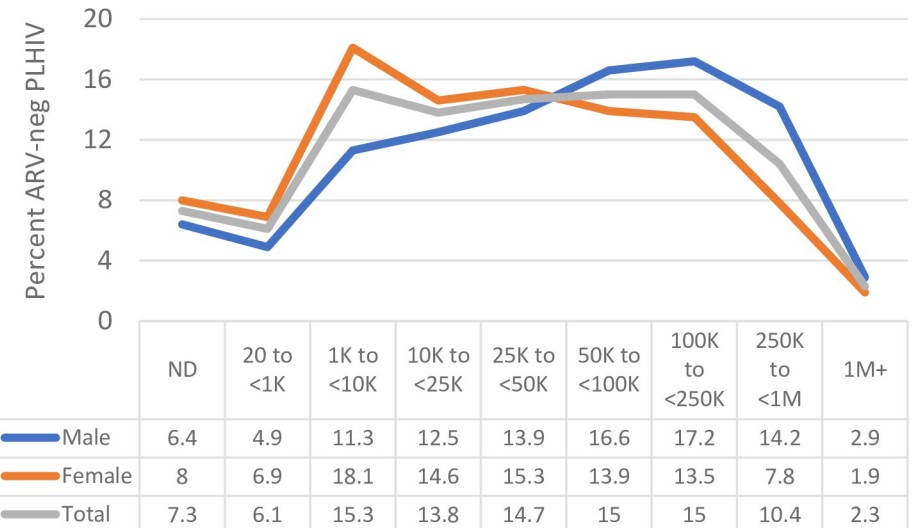

**Fig 1. Distribution of ARV-negative PLHIV by viral load category and by sex.** 12 Population-based HIV Impact Assessments, 2015–19. ARV: antiretroviral; PLHIV: people living with HIV; ND: Not detectable; K denotes values in thousands, M in millions.

proportions of suppressed PLHIV. The Gini coefficients (Table 1) ranged from 0.789 (Cote d'Ivoire) to 0.939 (Namibia and Rwanda). Drawing from the data informing the Lorenz curves, the proportion of the total VL corresponding to the 1% of PLHIV with the highest VL ranged from 17.1% (Cote d'Ivoire) to 47.2% (Rwanda); whereas the highest 5% of PLHIV accounted for 45.9% (Cote d'Ivoire) to 77.6% (Namibia).

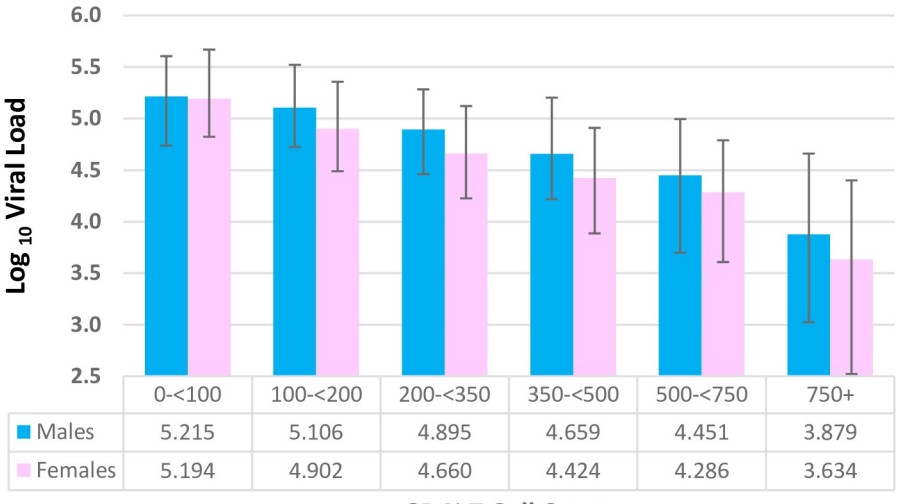

**Fig 2. Median log$_{10}$ viral load among ARV-negative PLHIV by CD4$^+$ T cell count category and sex.** 10 Population-based HIV Impact Assessments, 2015–19. Bars represent the 25th and 75th percentiles. ARV: antiretroviral; PLHIV: people living with HIV. Across all CD4 strata, the median viral load among males was 4.766; among females, 4.481.

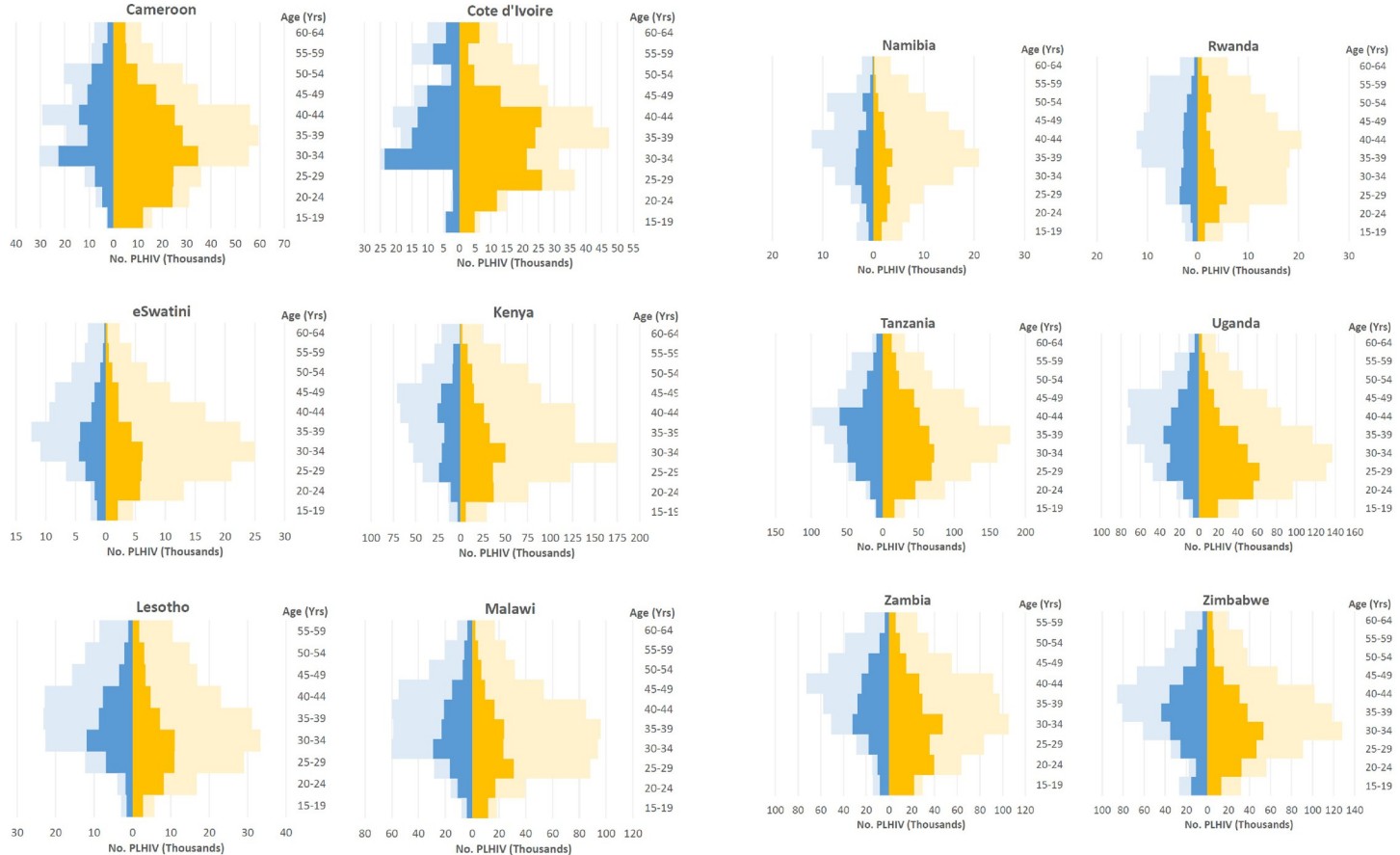

**Fig 3. Estimated number of suppressed and unsuppressed PLHIV by sex and age group.** Population-based HIV Impact Assessments, 2015–2019. PLHIV: (adult) people living with HIV. Dark blue color: Males unsuppressed, Light blue color: males suppressed, Dark orange color: females unsuppressed, Light orange color: females suppressed.

## Viremia, VL, and HIV incidence

Table 2 displays the HIV transmission ratios by country for both sexes combined as well as for opposite-sex transmission; corresponding measures of uncertainty are presented in S3 Table. Transmission ratios for both sexes combined ranged from 1.8 transmission events per 100 unsuppressed PY in Cote d'Ivoire to 15.0 in Uganda. In all countries, female-to-male transmission ratios were lower (ranging from 1.5 in Cote d'Ivoire to 10.6 in Uganda) than male-to-female transmission ratios (ranging from 2.3 in Cote d'Ivoire to 28.3 in Cameroon). Rearranging the data to estimate the number of unsuppressed adults per incident HIV per year suggests that across these 12 countries, in Uganda only 6.6 unsuppressed adults corresponded to each case of incident HIV whereas in Cote d'Ivoire the estimate was 56.1. We observe equally wide ranges when assuming that all transmission was between opposite sexes: For male-to-female transmission, country estimates range from 3.5 unsuppressed males per female incident HIV (Cameroon) to 44.1 in Cote d'Ivoire, whereas for female-to-male transmissions our estimates ranged from 9.4 unsuppressed females in Uganda to 67.4 in Cote d'Ivoire. Similar to transmission ratios, in all countries we estimate fewer unsuppressed males for each female incident HIV than vice versa.

Table 3 shows the correlation between select VL metrics and incident HIV. Correlations between population viremia and HIV incidence were strong (i.e., $R^2 > 0.70$) for both sexes

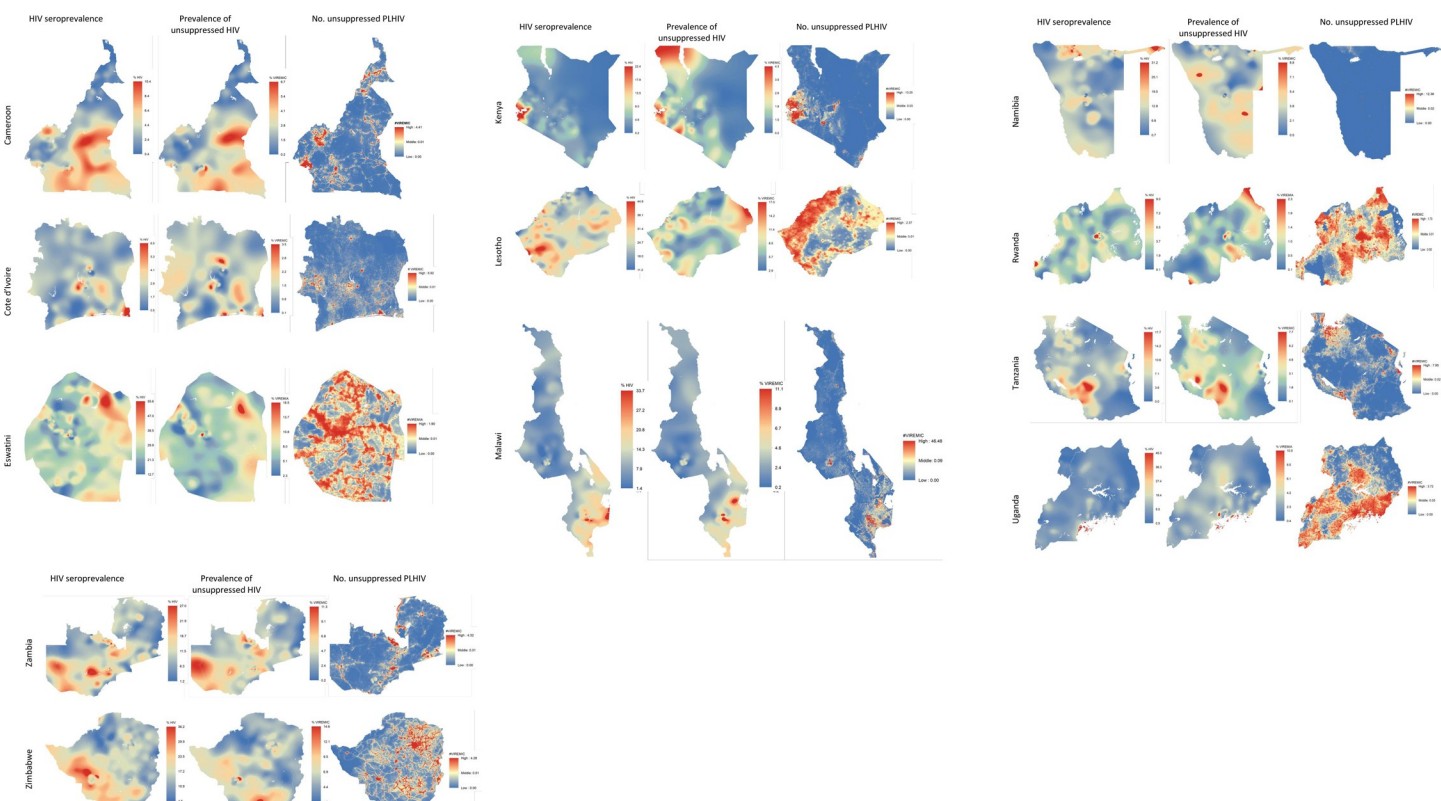

**Fig 4. Subnational HIV seroprevalence, viremia prevalence, and number unsuppressed PLHIV, 12 Population-based HIV Impact Assessments, 2015–19. Viremia prevalence** indicates the estimated percentage of the adult population (HIV-infected and uninfected) with unsuppressed viral load. **No. unsuppressed PLHIV** indicates the estimated number of adult unsuppressed PLHIV per pixel (100m by 100m square). Note: The coloring ramp may not correlate in linear fashion with the count values of unsuppressed PLHIV, and maps should be compared within, rather than across, countries.

combined, for opposite sexes, and between 30+ year-old unsuppressed males and 15–29-year-old HIV-incident females; only the correlation between younger unsuppressed females and older HIV-incident males was moderate (0.51). Gini coefficients also showed a strong correlation with an inverse relationship to incidence. A strong correlation was also seen with expressing a country's VL as total VL (absolute and $\log_{10}$) with the number of incident HIV infections for both sexes combined as well as between unsuppressed males and HIV-incident females. Whereas a modest correlation (i.e., between 0.50–0.70) was observed between unsuppressed females and HV-incident males. The correlations for both sexes combined weakened when restricting PLHIV to those diagnosed or on treatment. The number of unsuppressed PLHIV per 100 HIV-uninfected adults (No. unsuppressed/100 HIV-neg) as well as HIV seroprevalence showed the strongest correlations with HIV incidence ($R^2$ = 0.92).

## Discussion

We examined population VL in 12 SSA countries in an attempt to add to the understanding of HIV epidemiology in this region. Key observations include that female unsuppressed PLHIV continue to outnumber males while ART-naïve male PLHIV have consistently higher VL than females. HIV transmission ratios were higher for males than females and vary significantly geographically while the geospatial distribution of population viremia–for some countries—appears markedly different than that for HIV seroprevalence in select countries.

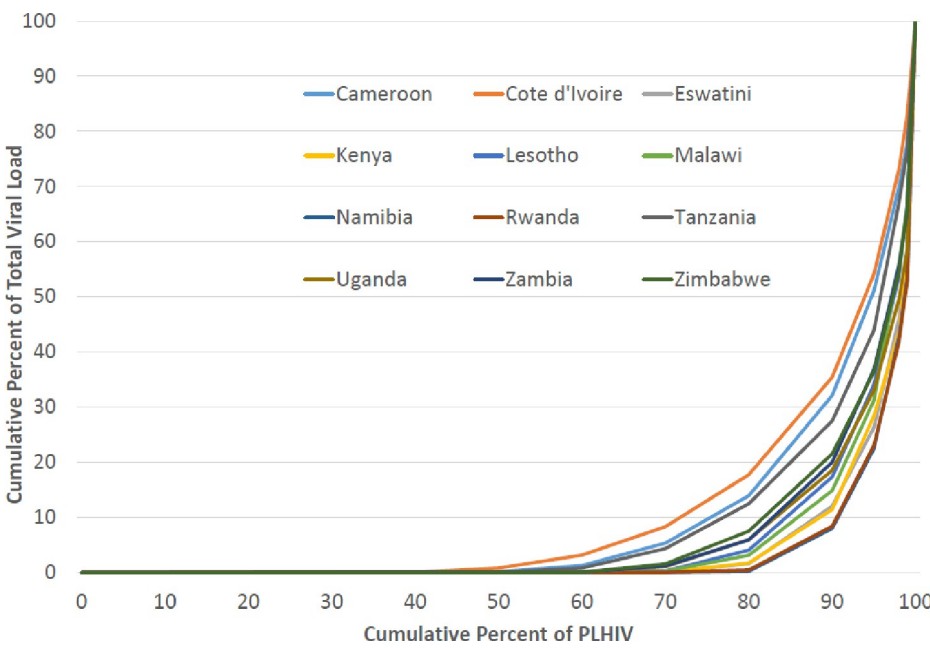

**Fig 5. Lorenz curves for the distribution of viral load.** 12 Population-based HIV Impact Assessments, 2015–2019. The x-axis displays the cumulative percent of all adult PLHIV ranked by viral load. The y-axis displays the cumulative percent of total VL. PLHIV: people living with HIV.

Our study benefitted from several strengths related to the PHIAs: We utilized standardized data from 12 large high-quality [31] national population-based surveys that allowed for the concurrent estimation of HIV incidence, prevalence, and VL-related metrics. The surveys' incidence estimates also inform UNAIDS-led national HIV incidence estimates and treatment status was measured through ARV testing. Nevertheless, several limitations need to be noted. Our surveys' antibody-based HIV testing algorithms could not detect acute HIV, a stage characterized by seronegativity and rapidly rising VL. We assumed that the overall level of viremia is long-term. We did not consider behavioral effects (e.g., partner change rates, condom use, male-to-male sex, sex work, or injection drug use) nor could we link HIV transmitting with HIV acquiring persons at the individual level. Further, we could not consider the contribution of HIV subtypes or HIV disease stage (primary, latent, late stage) to HIV transmission, although we did observe a few individuals with very high VL, suggesting primary HIV infection. Our estimated transmission ratios were accompanied by wide standard errors, mostly reflecting the small numbers of recent HIV infections detected in our surveys. Our surveys' cross-sectional design does not allow us to infer a causal relationship between viremia and incidence; as well, all our surveys' recent HIV infections were viremic as defined per our testing algorithm for recent HIV-1 infection.

As suggested in earlier work by others (e.g., [4, 32]) population viremia can be a more suitable indicator for monitoring HIV epidemics as it better reflects the current burden of disease as well as the remaining potential for HIV-related morbidity, mortality and onward transmission compared to conventional HIV prevalence estimates. Effective ART programs removed the bulk of these countries' total population VL, "concentrating" the remaining viremia in a shrinking subpopulation of unsuppressed PLHIV. Looking at the remaining prevalence of viremia rather than HIV seroprevalence, allowed us to paint a partially different picture of SSA's HIV epidemics, one where e.g., Lesotho and Eswatini no longer face a hyperendemic

**Table 2. Transmission metrics.** Population HIV Impact Assessments, sub-Saharan Africa, 2015–19.

| | Cameroon | Cote d'Ivoire | Eswatini | Kenya | Lesotho | Malawi | Namibia | Rwanda | Tanzania | Uganda | Zambia | Zimbabwe | Median |
|---|---|---|---|---|---|---|---|---|---|---|---|---|---|
| **HIV incidence** | | | | | | | | | | | | | |
| HIV incidence rate (per 100 HIV-neg PY) | 0.24 | 0.03 | 1.15 | 0.14 | 1.10 | 0.37 | 0.36 | 0.08 | 0.25 | 0.40 | 0.61 | 0.42 | 0.37 |
| No. incident infections | 31,376 | 4,054 | 5,717 | 35,922 | 9,835 | 27,973 | 4,468 | 5,419 | 72,124 | 72,619 | 42,855 | 29,194 | 28,584 |
| **HIV transmission ratio / 100 PLHIV / year** | | | | | | | | | | | | | |
| Both sexes | 6.3 | 1.1 | 3.0 | 2.8 | 3.2 | 3.1 | 2.5 | 2.6 | 4.8 | 6.1 | 4.5 | 2.5 | 3.0 |
| Male-to-female | 16.3 | 1.6 | 5.8 | 4.7 | 4.1 | 5.5 | 5.8 | 3.5 | 9.6 | 9.9 | 9.0 | 3.9 | 5.7 |
| Female-to-male | 1.8 | 0.8 | 1.5 | 1.9 | 2.6 | 1.6 | 0.7 | 2.1 | 2.4 | 3.9 | 1.7 | 1.6 | 1.8 |
| **HIV transmission ratio / 100 unsuppressed PLHIV / year** | | | | | | | | | | | | | |
| Both sexes | 11.3 | 1.8 | 10.9 | 9.7 | 9.9 | 9.8 | 11.2 | 10.7 | 10.1 | 15.0 | 11.2 | 6.3 | 10.4 |
| Male-to-female | 28.3 | 2.3 | 17.5 | 13.3 | 11.1 | 14.1 | 19.2 | 11.7 | 16.3 | 21.5 | 21.5 | 8.3 | 15.2 |
| Female-to-male | 3.2 | 1.5 | 6.3 | 7.4 | 8.9 | 5.9 | 3.9 | 10.0 | 5.7 | 10.6 | 4.5 | 4.5 | 5.8 |
| Male: female transmission rate ratio | 8.8 | 1.5 | 2.8 | 1.8 | 1.2 | 2.4 | 4.9 | 1.2 | 2.9 | 2.0 | 4.8 | 1.9 | 2.2 |
| **No. unsuppressed PLHIV per incident HIV** | | | | | | | | | | | | | |
| Both sexes | 8.8 | 56.1 | 9.1 | 10.3 | 10.1 | 10.2 | 8.9 | 9.3 | 9.9 | 6.6 | 8.9 | 15.9 | 9.8 |
| No. unsuppressed female PLHIV / male incident HIV | 30.9 | 67.4 | 15.8 | 13.5 | 11.2 | 16.9 | 25.4 | 10.0 | 17.7 | 9.4 | 22.4 | 22.3 | 17.3 |
| No. unsuppressed male PLHIV / female incident HIV | 3.5 | 44.1 | 5.7 | 7.5 | 9.0 | 7.1 | 5.2 | 8.5 | 6.1 | 4.7 | 4.6 | 12.0 | 6.6 |

**Note**: PLHIV: people living with HIV; PY: person-years. Transmission ratios indicate the estimated number of HIV transmissions (= incident HIV) for each 100 (unsuppressed) PLHIV per year.

epidemic (i.e., <15% viremia prevalence among adults [33]), Rwanda no longer has a generalized epidemic (<1%) and Kenya (1.4%) is close to achieving the same milestone.

Similarly, our viremia-based maps offer a more informative display of the spatial distribution of viremia compared to HIV seroprevalence. In some countries, the spatial distribution of viremia prevalence simply reflected the spatial distribution of HIV infection (Cote d'Ivoire, Malawi, Uganda). In other countries (Kenya, Cameroon, Namibia) clear differences emerged, apparent HIV "hotspots" subsided and "viremia hotspots" surfaced as distinct from HIV prevalence hotspots, illustrating areas with poor viral suppression. Other country comparisons were more mixed, nuanced, and intermediate, showing some areas with varying viral suppression. Our third type of maps, displaying spatial counts of unsuppressed PLHIV, often differed markedly from the two proportional displays of HIV and viremia. These count maps are a function of population density, HIV prevalence, and the suppressive effect of ART programs on VL. It is likely this latter type of map that may prove most useful for program planning and evaluation as it reflects the absolute burden of remaining unsuppressed HIV. In contrast, HIV seroprevalence and prevalence of viremia do not inform about the absolute burden (i.e., number of people with) of HIV or unsuppressed HIV, respectively. Despite the surveys' large sample sizes, the number of detected participants with recent HIV infections was too small to meaningfully examine the spatial distribution of recent HIV infection alongside viremia.

A population's mean and sum total VL are rarely used metrics reflecting the absolute size of its transmission and morbidity potential [7, 8] and was shown to correlate with the number of new HIV diagnoses [9]. In our study, the 12 countries' total VL varied 19-fold, a function of population size, number of PLHIV, and ART uptake, and was strongly correlated with HIV incidence (both sexes), albeit much less so when restricted to female total VL.

**Table 3. Correlation between select HIV metrics and incident HIV.** 12 Population-based HIV Impact Assessments, 2015–19.

| Population viremia | HIV incidence rate | $R^2$ |
|---|---|---|
| Both sexes | Both sexes | 0.8896 |
| Females | Males | 0.8337 |
| Males | Females | 0.7869 |
| Females 15–29 yrs | Males 30+ yrs | 0.5063 |
| Males 30+ yrs | Females 15–29 yrs | 0.8027 |
| **No. unsuppressed/100 HIV-neg** | **HIV incidence rate** | $R^2$ |
| Both sexes | Both sexes | 0.9204 |
| **Gini coefficient** | **HIV incidence rate** | $R^2$ |
| Both sexes (all adults) | Both sexes | 0.8475 |
| **Mean VL** | **HIV incidence rate** | $R^2$ |
| Both sexes (all adults) | Both sexes | 0.5680 |
| **HIV seroprevalence** | **HIV incidence rate** | |
| Both sexes | Both sexes | 0.9209 |
| Females | Males | 0.8237 |
| Males | Females | 0.8555 |
| **Total VL** | **No. incident HIV** | $R^2$ |
| *Billions copies/mL* | | |
| All PLHIV | Both sexes | 0.8373 |
| Females | Males | 0.5325 |
| Males | Females | 0.8516 |
| Diagnosed PLHIV | Both sexes | 0.5790 |
| On ART PLHIV | Both sexes | 0.5442 |
| *$Log_{10}$ VL* | | |
| Both sexes | Both sexes | 0.8121 |
| Females | Males | 0.6796 |
| Males | Females | 0.8222 |

**Note**: HIV: human immunodeficiency virus; $Log_{10}$: log base 10; mL: milliliter; PLHIV: people living with HIV; $R^2$: coefficient of correlation; VL: viral load; yrs: years. $R^2$ values should not be interpreted as a causal relationship between VL metrics and incident HIV. Population viremia was computed as the proportion of all adults with unsuppressed VL. No. unsuppressed/100 HIV-neg was computed as the number of unsuppressed adults per 100 HIV-uninfected adults. Gini coefficients and mean VL were computed using all adults (HIV-infected and uninfected). Total VL represents the sum total VL of all adult PLHIV.

The $log_{10}$ transformed mean VL among unsuppressed PLHIV varied substantially across countries. We did not examine possible factors such as HIV subtypes or the prevalence of co-infections (e.g., herpes simplex virus type 2) [34] for this variation. The literature suggests that the per act HIV transmission risk is closely correlated to $log_{10}$ plasma VL [14] which may help explain some of the variation for HIV incidence across countries. Of note, unsuppressed males in all countries showed higher mean $log_{10}$ VL values compared to unsuppressed females, an observation made previously, albeit in smaller settings [35]. This is consistent with our esti-mated transmission ratios which in all countries were larger for male-to-female than for female-to-male transmission although without doubt other behavioral and biologic factors [36] disproportionately contribute to female HIV acquisition as well. The highest mean $log_{10}$ VL (5.15) was estimated for unsuppressed males in Cameroon, the same country for which we also estimated the highest male-to-female transmission ratio (28.3).

Although females generally achieve higher proportions of VLS compared to males in all PHIA countries, examining the absolute numbers of unsuppressed PLHIV suggests that in all countries unsuppressed females still outnumber unsuppressed males, due to the higher HIV prevalence observed among females. This finding may serve as a reminder that while more progress towards VLS among males is warranted, most of the absolute unsuppressed HIV burden in SSA continues to be shouldered by females. These data also suggest that estimating the number of unsuppressed PLHIV across demographic and behavioral strata has its own merit, beyond VLS monitoring.

Lorenz curves, commonly used to examine income inequality, are an uncommon way to examine VL distribution [10]. While perhaps less useful or needed for routine epidemic monitoring, the extreme disparities depicted in our Lorenz curves illustrate the large-scale successes of most national ART programs that resulted in VLS in many or most PLHIV. As a result, at the time of these surveys, the 1% of adult PLHIV with the highest VL on average accounted for one third of countries' remaining total VL, and the highest 5% accounted for two-thirds of total VL. Identifying, linking, and improving adherence of this small population segment to ART programs would suggest disproportional gains in reducing total VL and the corresponding HIV transmission potential, albeit our data did not allow us to infer that PLHIV with (very) high VL are also at higher transmission risk behaviorally. The corresponding Gini coefficients' large values (close to 1) express these extreme but desired inequalities numerically and display a strong correlation with HIV incidence.

Examining VLS against ARV status showed that between 9% and 21% of PLHIV who tested ARV-negative were nevertheless suppressed. Given SSA's public health approach to ART provision, we believe that few if any of our survey respondents were taking ARVs that our testing did not cover. These rather large proportions may be due to short-term lapses in ARV intake among some participants whereas a minority may have been elite controllers [37] or transiently controlled VL [38]. Conversely, 6% to 21% of PLHIV who tested ARV-positive were unsuppressed, indicating poor adherence, drug resistance, or having recently initiated ART. These are sizeable proportions that call for further improvement in adherence and drug resistance monitoring.

Our estimated HIV transmission ratios are similar to those reported for the US [39]. Transmission ratios and number of unsuppressed adults per incident case of HIV highlight a very different scale compared to HIV incidence rates. Whereas incidence rates contain the entire HIV-uninfected population in its denominator (at risk for HIV acquisition), effective transmission ratios indicate the force of HIV transmission from the perspective of the much smaller population of unsuppressed adults (at risk for HIV transmission). HIV transmission ratios may be seen as aggregate estimates of annualized effective reproductive numbers. The much higher values for effective transmission ratios (compared to incidence rates) underscore the importance of this relatively small sub-population of unsuppressed adults for ongoing HIV transmission risk. Most countries showed somewhat similar effective transmission ratios (combined for both sexes) between approximately 10 to 11 HIV transmission events per 100 unsuppressed adults per year; outliers included Uganda (15.0), Zimbabwe (6.3) and Cote d'Ivoire (1.8). Looking at the opposite sex transmission ratios we note two general observations: The male-to-female transmission ratios were uniformly higher than female-to-male transmission ratios and the sex-specific variations across countries were more marked. Cameroon and Cote d'Ivoire specifically warrant mention: Cote d'Ivoire offered the lowest estimated HIV transmission ratios, whereas Cameroon saw the largest difference between male-to-female and female-to-male transmission ratios. The reasons for the marked variations for sex-specific transmission ratios across countries and the extremely high male-to-female effective transmission ratio in Cameroon remain unclear and are beyond the scope of this analysis.

Data from the same surveys suggest that the self-reported prevalence of medical male circumcision varied from 9.8% (Malawi) to 58.1% (Cameroon) [17].

Our estimated transmission ratios assume that all transmission occurred 1) within the respective countries, 2) within the sampled adult age ranges (no transmission between HIV-infected children and adults, or, where older age groups remained unsampled, from older age groups to sampled adult age groups), and 3) for sex-specific ratios, between opposite sexes (i.e., male-to-male transmission was ignored). We believe these assumptions are acceptable for entire countries (compared to smaller, subnational studies) although we recognize that our female-to-male transmission ratios are overestimates as some male-to-male transmission will have contributed to the estimated male HIV incidence (UNAIDS estimates 6% and 21% of incident HIV in 2020 in Eastern/Southern and Western/Central Africa was due to male-to-male sex, respectively [40]). To the extent that our estimates for HIV incidence and population viremia are national-level estimates, the resulting HIV transmission ratios incorporate most/all incident HIV occurring inside and outside of stable sexual relationships. We could not account for the potential HIV transmission between sex partners residing in different countries, although the net effect (offsetting 'exported' transmissions by 'imported' ones) can be expected to be somewhat smaller than the absolute number of such transmissions. Even so, our surveys sampled all 'de-facto' household members and visitors who slept in the sampled dwelling the night before, regardless of nationality. For the same reasons, estimates of transmission ratios quickly become unreliable when restricted to subgroups, such as age (e.g., young females and older men). Unlike HIV acquisition, no detectable signal can reliably identify transmitting individuals (except for the use of phylogenetic analysis).

Across the 12 PHIA countries, we were able to confirm a strong correlation between population VL and incident HIV, as confirmed in previous studies elsewhere [4, 32]. The substantially higher $R^2$ values for transmission from older unsuppressed males to younger HIV-incident females (compared to the opposite scenario) is supported by other studies [41, 42] and highlights again a potentially important component in SSA's HIV epidemics. Expressing total VL in absolute or $\log_{10}$ values showed no discernible difference. The lower $R^2$ values shown when restricting the population to PLHIV who are diagnosed or on treatment serve as a reminder about the advantage of population-based data as compared to routine service data that by definition are restricted to PLHIV utilizing such services [4].

With the large-scale roll-out of ART in SSA, the declining utility of HIV prevalence to monitor the epidemic may be compensated using VL. Given that virtually all transmission stems from unsuppressed individuals, viremia-related metrics indicate a population's HIV transmission potential, can be estimated with more precision than HIV incidence in the survey or surveillance setting, and are more amenable to programmatic action. As a continuous or categorical metric, VL can inform various indicators for surveillance, evaluation, and monitoring purposes and are an excellent way to describe the spatial and demographic distribution of HIV-related transmission, morbidity, and mortality risk. The population-level prevalence of unsuppressed HIV and the number of unsuppressed PLHIV, disaggregated as needed, stand out in their added utility for epidemic monitoring and HIV programming and should be reported and evaluated whenever feasible.

## Supporting information

**S1 Table. Adult population, PLHIV, and number unsuppressed at time of survey and adjustments in creation of raster-based population counts.**
(DOCX)

**S2 Table. Standard errors and 95% confidence intervals for Table 1.**
(XLSX)

**S3 Table. Standard errors and 95% confidence intervals for transmission ratios shown in Table 2.**
(XLSX)

**S1 File. IRB approval documentation.**
(DOCX)

**S2 File. PLOS questionnaire on inclusivity.**
(DOCX)

## Acknowledgments

We thank the PHIA survey respondents for their participation and PHIA study staff for their work. We are grateful for the many constructive comments received from the reviewers of this manuscript.

**Disclaimer:** The findings and conclusions in this report are those of the author(s) and do not necessarily represent the official position of the U.S. Department of Health and Human Services, the Centers for Disease Control and Prevention.

## Author Contributions

**Conceptualization:** Wolfgang Hladik, Paul Stupp.

**Data curation:** Wolfgang Hladik, Paul Stupp, Jessica Justman, Clement Ndongmo, Judith Shang, Emily K. Dokubo, Elizabeth Gummerson, Isabelle Koui, Stephane Bodika, Roger Lobognon, Hermann Brou, Caroline Ryan, Kristin Brown, Harriet Nuwagaba-Biribonwoha, Leonard Kingwara, Peter Young, Megan Bronson, Duncan Chege, Optatus Malewo, Yohannes Mengistu, Frederix Koen, Andreas Jahn, Andrew Auld, Sasi Jonnalagadda, Elizabeth Radin, Ndapewa Hamunime, Daniel B. Williams, Eugenie Kayirangwa, Veronicah Mugisha, Rennatus Mdodo, Stephen Delgado, Wilford Kirungi, Lisa Nelson, Christine West, Samuel Biraro, Kumbutso Dzekedzeke, Danielle Barradas, Owen Mugurungi, Shirish Balachandra, Peter H. Kilmarx, Godfrey Musuka, Hetal Patel, Bharat Parekh, Katrina Sleeman, Robert A. Domaoal, George Rutherford, Tsietso Motsoane, Anne-Cécile Zoung-Kanyi Bissek, Mansoor Farahani, Andrew C. Voetsch.

**Formal analysis:** Paul Stupp.

**Investigation:** Wolfgang Hladik, Stephen D. McCracken.

**Methodology:** Wolfgang Hladik, Stephen D. McCracken.

**Project administration:** Andrew C. Voetsch.

**Validation:** Paul Stupp.

**Visualization:** Wolfgang Hladik, Paul Stupp, Stephen D. McCracken.

**Writing – original draft:** Wolfgang Hladik, Stephen D. McCracken.

**Writing – review & editing:** Wolfgang Hladik, Jessica Justman, Clement Ndongmo, Judith Shang, Emily K. Dokubo, Elizabeth Gummerson, Isabelle Koui, Stephane Bodika, Roger Lobognon, Hermann Brou, Caroline Ryan, Kristin Brown, Harriet Nuwagaba-Biribonwoha, Leonard Kingwara, Peter Young, Megan Bronson, Duncan Chege, Optatus

Malewo, Yohannes Mengistu, Frederix Koen, Andreas Jahn, Andrew Auld, Sasi Jonnalagadda, Elizabeth Radin, Ndapewa Hamunime, Daniel B. Williams, Eugenie Kayirangwa, Veronicah Mugisha, Rennatus Mdodo, Stephen Delgado, Wilford Kirungi, Lisa Nelson, Christine West, Samuel Biraro, Kumbutso Dzekedzeke, Danielle Barradas, Owen Mugurungi, Shirish Balachandra, Peter H. Kilmarx, Godfrey Musuka, Hetal Patel, Bharat Parekh, Katrina Sleeman, Robert A. Domaoal, George Rutherford, Tsietso Motsoane, Anne-Cécile Zoung-Kanyi Bissek, Mansoor Farahani, Andrew C. Voetsch.

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
