## [Decision Letter · Decision Letter 0]

25 Nov 2022

PONE-D-22-26006The epidemiology of HIV population viral load in twelve sub-Saharan African countriesPLOS ONE

Dear Dr. Hladik,

Thank you for submitting your manuscript to PLOS ONE. After careful consideration, we feel that it has merit but does not fully meet PLOS ONE’s publication criteria as it currently stands. Therefore, we invite you to submit a revised version of the manuscript that addresses the points raised during the review process.

We look forward to receiving your revised manuscript.

Kind regards,

José Antonio Ortega, Ph.D.

Academic Editor

PLOS ONE

Journal Requirements:

2. Please amend your current ethics statement to include the full names of all ethics committees/institutional review boards that approved your specific study.

3. Please include your ethics statement in the Methods section of your manuscript. In the Methods section of your revised manuscript, please include the full name of the institutional review boards or ethics committees that approved the protocol, the approval or permit number that was issued, and the date that approval was granted.

4. Please include a complete copy of PLOS’ questionnaire on inclusivity in global research in your revised manuscript. Our policy for research in this area aims to improve transparency in the reporting of research performed outside of researchers’ own country or community. The policy applies to researchers who have travelled to a different country to conduct research, research with Indigenous populations or their lands, and research on cultural artefacts. The questionnaire can also be requested at the journal’s discretion for any other submissions, even if these conditions are not met.  Please find more information on the policy and a link to download a blank copy of the questionnaire here: https://journals.plos.org/plosone/s/best-practices-in-research-reporting. Please upload a completed version of your questionnaire as Supporting Information when you resubmit your manuscript.

5. You indicated that you had ethical approval for your study. In your Methods section, please ensure you have also stated whether you obtained consent from parents or guardians of the minors (participants under the age of 18 years) included in the study or whether the research ethics committee or IRB specifically waived the need for their consent.

6. We note that the grant information you provided in the ‘Funding Information’ and ‘Financial Disclosure’ sections do not match. 

7. We note that you have included the phrase “data not shown” in your manuscript. Unfortunately, this does not meet our data sharing requirements. PLOS does not permit references to inaccessible data. We require that authors provide all relevant data within the paper, Supporting Information files, or in an acceptable, public repository. Please add a citation to support this phrase or upload the data that corresponds with these findings to a stable repository (such as Figshare or Dryad) and provide and URLs, DOIs, or accession numbers that may be used to access these data. Or, if the data are not a core part of the research being presented in your study, we ask that you remove the phrase that refers to these data.

8. Please include your full ethics statement in the ‘Methods’ section of your manuscript file. In your statement, please include the full name of the IRB or ethics committee who approved or waived your study, as well as whether or not you obtained informed written or verbal consent. If consent was waived for your study, please include this information in your statement as well. 

Additional Editor Comments:

Three reviewers have evaluated the manuscript providing valuable suggestions for improvement. I would qualify overall the comments suggested as minor changes since they essentially involve minor editing, improved motivation / explanation, and formatting of tables. In that respect you should make sure to follow PLOS ONE guidelines for statistical reporting. If you believe that in terms of readability it is better to focus on a subset of the information items, you can make use of appendix tables that are comprehensive.

Reviewer 1 raises particular suggestions on the ordering of authors from African countries. Note that you do not need to follow this suggestion since PLOS ONE does not have a specific policy on the ordering of authors in collaborative research. The guidelines for authorship are PLOS criteria for authorship. Note, however, that PLOS ONE is concerned about inclusion in global research and that you will be contacted by the journal to fill a questionnaire on inclusivity. The information collected in the questionnaire will be made available to the editor and reviewers during the peer review process to help assess whether the research meets the journal’s standards for the ethics of experimentation and research integrity. The completed questionnaire would also be included as a Supporting Information file with the published paper, improving transparency in global research.

Reviewer 2 suggests focusing only on a subset of indicators. You can decide on that. If you believe it is useful to provide an overview of the survey results, this is acceptable, but you have to make sure that enough explanation is included for a general audience to understand what is being analyzed and why is it relevant, including the relevant literature.

Reviewers' comments:

Reviewer's Responses to Questions

**Comments to the Author**

1. Is the manuscript technically sound, and do the data support the conclusions?

Reviewer #1: Yes

Reviewer #2: Yes

Reviewer #3: Yes

2. Has the statistical analysis been performed appropriately and rigorously? 

Reviewer #1: Yes

Reviewer #2: No

Reviewer #3: Yes

3. Have the authors made all data underlying the findings in their manuscript fully available?

Reviewer #1: Yes

Reviewer #2: Yes

Reviewer #3: Yes

4. Is the manuscript presented in an intelligible fashion and written in standard English?

Reviewer #1: Yes

Reviewer #2: Yes

Reviewer #3: Yes

5. Review Comments to the Author

Reviewer #1: The manuscript by Dr Hladik and colleagues reports on the epidemiology of HIV population viral load in twelve sub-Saharan African countries. The authors used data generated in the framework of Population-based HIV Impact Assessments (PHIAs) conducted between 2015 and 2019 to perform their analyses. These were well performed, with appropriate methods. The manuscript, although very long and dense, is well written. The findings presented are important at both country and international levels. The authors are thus commended for their endeavor.

The work deserves some comments.

Major

1. The major comment I have is that the work was performed in 12 sub-Saharan African countries but the first 7 and the last two authors are from outside Africa. This is not normal and the authors should change that before any other further consideration.

Minor

1. On page 8, line 155, it is written that HIV-2 data were excluded from the analysis. Can the authors indicate how many of such cases were recorded, presumably all from Côte d’Ivoire?

2. On page 4, line 68in the introduction, it is written that VLS eliminates the risk of HIV transmission. The world eliminates is too strong. Maybe “reduces” or “minimizes” as written line 71 of the same page is more appropriate.

3. One factor that was not considered in the analysis in the 12 countries is HIV-1 diversity as explaining factor of observed differences. Between countries of Wes-Central Africa, those of Easter Africa and finally those of Southern Africa, different HIV-1 subtypes are circulating, which slightly differ in their fitness, susceptibility to ART and transmissibility. The authors should mention in the discussion these possibilities also.

4. In the discussion section, the authors mentioned male circumcision (page 26-27). This parameter is not the same in West and Central Africa, where most of the males are circumcised, and in the Southern Africa. I think giving at least the proportion of circumcised males in different countries remain an interesting information to share.

5. Finally, I find the discussion quite long.

Reviewer #2: I make this review in my personal capacity and not as a staff of UNAIDS.

The authors have worked hard, demonstrating a strong grasp of the topic. However, the paper is not ready for publication.

With major revisions this paper may make an important contribution to the body of knowledge on viral load metrics.

The questions being investigated and why they are being investigated need clear explanation, and some of the main literature a closer read, to determine if the approaches undertaken are valid.

For example, where does 0 start for the Lorenzo curve. Is it at less than 1K HIV viral load, since undetectable does not contribute to the total viral load? Has the viral load measurement been lognormalised, or scaled in way that is easier to plot? Further, it is not clear if a viral load of 3000 is better than a viral load of 4000 or 5000 in terms of HIV transmission dynamics. For this reason, the lorenz curve and Ginie co-efficient may not be an appropriate method to measure inequality in viral load suppression because HIV viral load does not have the same properties as income or wealth. Maybe remove the lorenz curve analysis.

And also see which other analysis you may remove to improve the clarity of the paper. The metrics although few are each complex and not easy to follow. Focusing on one of two metrics may improve the usefulness of the analysis.

Table 2 shows promise if its better explained and standardized. For example, the standard errors are only included in the HIV transmission viremia and not in others. And also, the standard errors are large and there is no discussion of these standard errors, if they robust standard errors. Maybe remove the standard errors.

Overall, the tables, graphs and maps are well done. They would be clear if the axis are labelled and each table or figure briefly described before presenting the results. If possible, put the description of each table or figure on the same page to help the reviewer follow what each figure or table is conveying, depending on what the journal requires.

Figure 1 is not clear what is being plotted, 20 is less than 1k in the Table so the header of second column should be 1K. Similarly figure 20 is less than 100.

For the maps indicate the units of measurements as you did for the population graphs e.g. 100s or 1,000S. Also see if you want to present both prevalence and viremia because the information appear not too different between the two, after addressing the earlier comments on calculating population viremia.

Please see the following points that may help improve the paper.

Include the data set from Ethiopia since it is available. If not indicate why you excluded the data set from Ethiopia. Because it was conducted only in urban areas may not be sufficient reason to exclude it.

Change female to women, male to men after the first mention, through out the paper.

Include the names of the country of the sample size. I did not see which country has a sample size of 30,637 indicated in the abstract. Please check.

"The number of viremic women outnumbered the viremic men in all countries" Is an important finding. Please present this result also in proportions and devote maybe a paragraph or so discussing the results and its policy implications.

On page 4, please read carefully the new UNAIDS Strategy and paraphrase from this strategy information on line 66 - 68 to something like "focuses on eliminating inequalities in access to HIV services...." It may not be correct to present the strategy as focusing on treatment as prevention.

Also, on line 71 include to be met by 2025 next to the 95-95-95.

Line 77 "... correlated with (add) HIGHER..."

Line 80, please rephrase the sentence after the sources, becuase dichotomizing VL is useful as an analytic variable. It helps policy makers to track viral load suppression

Line 155 please explain why this was excluded.

On line 161 include the laboratory confirmation of results.

Line 171. It may not be correct to multiply the HIV prevalence with the proportion of people who were unsuppressed. With this approach countries such as Eswatini with higher HIV prevalence will have lower population HIV viral load, for example, for the same proportion of people with unsuppressed viral loads.

Kindly add the total of values unsuppressed viral load to find the total national value of HIV viral load that is not suppressed.

On page 12, I don't see HIV incidence data. Kindly clarify.

On page 14, the statement "Except for CD4 T cell counts below..." may not true because the confidence intervals overlap. Also presenting the confidence intervals as 25% and 75% is not easy to follow, especially as these demarcations are not discussed in the discussion section. Please revert to the 95% CI or 90% CI, if not able to discuss in details the 25% and 75%.

Page 20, Table 3, analyzing correlation between population viremia and HIV incidence is not correct. HIV incidence is in population viremia. So you are measuring the same thing. If you want to go further with this metric maybe use another method such as instrumental variables. Maybe best to remove this table.

On page 22, second sentence regarding the PHIA sampling PLHIV should be moved to methods section. Similarly, the sentence after source 30, should be moved to the methods section.

Page 24 and 26. Cameroun and Côte D'Ivoire's sample size are generally smaller. Analysis focusing on these countries need additional investigations.

Page 25, first paragraph is very important and some effort should be spent on discussing this finding.

This is an important topic which with further revisions may add a lot to the body of knowledge on viral load metrics. I am happy to review further the revised manuscript.

Reviewer #3: Reviewer Report

The epidemiology of HIV population viral load in twelve sub-Saharan African countries

The topic is relevant and important metrics were highlighted. The analysis are intense and well executed. However, was looking forward to seeing data from country like South Africa, which is the epicentre of HIV in sub-Saharan Africa. Not sure why the authors did not include it.

Introduction

Line 70 -71: The reference provided was for the 90-90-90 target, the revised 95-95-95 target needs to be referenced

Material and methods

Line 97: Firstly, I suggest the subheading to read “Data Sources, Setting and Study design” instead of Setting and study design. Secondly the sub section did not clearly show the source of data but more of the study sponsors. Thirdly, a few more description of the study design is needful.

Line 99: Grammar error. response efforts in “select low- and middle-income countries”, … Should be …in selected low- and middle-income countries …….

Line 101: Grammar error. Implementing partners included country governments, grammar error …. should be …. “Implementing partners includes country governments

Line 112: “The proportion of survey participants who consented to a blood draw” …… Relook at the sentence, need to be rephrased

Line 108 -111: Shows various data used and number. Why was South Africa data not included. Was it intentional or was it not available.

However, from the introduction line 90-91, it was stated that data are becoming increasingly available with South Africa in reference. It will be good to include South Africa data in the analysis otherwise, the authors can remove South Africa from Line 91. As not having data from that country in the analysis make the statement contradicting.

Including South Africa data in the analysis is highly recommended. Good to see the correlation of the VL metrics from other countries and South Africa, especially Eswatini

Line 113: The gender inclusion should be reflected in the participants.

Line 222. …….by including all adults …. kindly state the age brackets for the adults.

Line 226: References or the ethical approval numbers from each country should follow the statement in this line.

Data management and analysis

Line 196 to 199: The Gini coefficient and Lorenz curves were defined/ explained. Kindly Provide Reference(s). Furthermore. A brief explanation of the correlation range is needful. When do we say the correlation is strong, weak, or moderate (show by values)

Results

Line 235: Check grammar …. Table 1 shows select HIV and VL- related metrics. This should be ……Table 1 shows selected HIV and ……….

There was NO NUMBER LINE after 246 in the Results and Discussion sections. Thus, referring to any review was impossible. Adding number line to these sections is then recommended to enable further review.

Tables and Figures looks good

Supplementary material

Figures in the Supplementary material should be relabelled as Figure S1, S2, ,,,,,, for clear distinction from the main document figures

6. PLOS authors have the option to publish the peer review history of their article (what does this mean?). If published, this will include your full peer review and any attached files.

Reviewer #1: No

Reviewer #2: **Yes: **David Chipanta

Reviewer #3: **Yes: **Adenike .O. Soogun

---

## [Author Response · Author response to Decision Letter 0]

30 Jan 2023

We uploaded our responses to reviewer comments as a separate rebuttal letter.

---

## [Decision Letter · Decision Letter 1]

13 Feb 2023

PONE-D-22-26006R1The epidemiology of HIV population viral load in twelve sub-Saharan African countriesPLOS ONE

Dear Dr. Hladik,

Thank you for submitting your manuscript to PLOS ONE. After careful consideration, we feel that it has merit but does not fully meet PLOS ONE’s publication criteria as it currently stands. Therefore, we invite you to submit a revised version of the manuscript that addresses the points raised during the review process.

The three previous experts have provided their input on the revised version. Two of them are satisfied with the changes made. Note that, in the case of referee 1 who keeps the previous reservations, there is no comment on the filled inclusivity questionnaire where you explain the rationale behind the author's list.Regarding the comments of referee 2, please be advised that providing an idea of uncertainty is important in any work based on a sample survey. If CIs are removed from the figures for clarity, you can include them in an appendix table/figure. You can also displace slightly the error bars in fig. 2 with position.dodge or a similar argument so that all error bars can be seen.

We look forward to receiving your revised manuscript.

Kind regards,

José Antonio Ortega, Ph.D.

Academic Editor

PLOS ONE

Journal Requirements:

Reviewers' comments:

Reviewer's Responses to Questions

**Comments to the Author**

1. If the authors have adequately addressed your comments raised in a previous round of review and you feel that this manuscript is now acceptable for publication, you may indicate that here to bypass the “Comments to the Author” section, enter your conflict of interest statement in the “Confidential to Editor” section, and submit your "Accept" recommendation.

Reviewer #1: All comments have been addressed

Reviewer #2: (No Response)

Reviewer #3: All comments have been addressed

2. Is the manuscript technically sound, and do the data support the conclusions?

Reviewer #1: Yes

Reviewer #2: Yes

Reviewer #3: Yes

3. Has the statistical analysis been performed appropriately and rigorously? 

Reviewer #1: Yes

Reviewer #2: No

Reviewer #3: Yes

4. Have the authors made all data underlying the findings in their manuscript fully available?

Reviewer #1: Yes

Reviewer #2: Yes

Reviewer #3: Yes

5. Is the manuscript presented in an intelligible fashion and written in standard English?

Reviewer #1: Yes

Reviewer #2: Yes

Reviewer #3: Yes

6. Review Comments to the Author

Reviewer #1: Most of my queries have been addressed by the authors. All the scientific questions have been addressed appropriately. The answer provided to my major comment is not satisfactory from an ethical point of view.

Reviewer #2: Many thanks for this useful paper and the hard work that has gone into it. The paper needs some minor revision in order for it to be ready for publication. Several issues need to be resolved: first the correlation between population viremia and HIV incidence is problematic because of the potential for reverse causality between population viremia and HIV incidence. Some of the discussions are tangental and can be taken out without losing much. I like that you recognise that the ratios are important but by themselves do not say much and need additional information to make them useful. The estimates need confidence intervals. I am sure you have this information.

Some of the information on the first page relating to the UNAIDS strategy and others are not correct. Please see how to rephrase.

Reviewer #3: Recommended for publication

7. PLOS authors have the option to publish the peer review history of their article (what does this mean?). If published, this will include your full peer review and any attached files.

Reviewer #1: No

Reviewer #2: No

Reviewer #3: **Yes: **Adenike O Soogun

---

## [Author Response · Author response to Decision Letter 1]

30 Mar 2023

Please see our updated Cover Letter and new Rebuttal, thank you! Wolfgang Hladik

---

## [Decision Letter · Decision Letter 2]

27 Apr 2023

PONE-D-22-26006R2The epidemiology of HIV population viral load in twelve sub-Saharan African countriesPLOS ONE

Dear Dr. Hladik,

Thank you for submitting your manuscript to PLOS ONE. After careful consideration, we feel that it has merit but does not fully meet PLOS ONE’s publication criteria as it currently stands. Therefore, we invite you to submit a revised version of the manuscript that addresses the points raised during the review process. Of the three previous reviewers, reviewer 2 still had some concerns and provides detailed suggestions together with reasoning for its convenience. You should either adopt the changes or argue on the convenience of not doing so. Regarding the inclusion of uncertainty estimates, it has improved the paper. Note that confidence intervals are often very wide. Some of reviewer 2 suggestions go in the direction of making sure that the claims in the paper take into account this uncertainty, and this is important throughout the manuscript. On a more specific detail, in figure 2 the lines overlap. Think of using an option, such as interval dodge in R, so that all the lines and intervals are visible.

We look forward to receiving your revised manuscript.

Kind regards,

José Antonio Ortega, Ph.D.

Academic Editor

PLOS ONE

Journal Requirements:

Reviewers' comments:

Reviewer's Responses to Questions

**Comments to the Author**

1. If the authors have adequately addressed your comments raised in a previous round of review and you feel that this manuscript is now acceptable for publication, you may indicate that here to bypass the “Comments to the Author” section, enter your conflict of interest statement in the “Confidential to Editor” section, and submit your "Accept" recommendation.

Reviewer #2: (No Response)

2. Is the manuscript technically sound, and do the data support the conclusions?

Reviewer #2: Yes

3. Has the statistical analysis been performed appropriately and rigorously? 

Reviewer #2: Yes

4. Have the authors made all data underlying the findings in their manuscript fully available?

Reviewer #2: Yes

5. Is the manuscript presented in an intelligible fashion and written in standard English?

Reviewer #2: Yes

6. Review Comments to the Author

Reviewer #2: My main issues on the potential for reverse causality between population HIV viral load and HIV incidence has not been addressed. Please also a few minor issues included in the review.

7. PLOS authors have the option to publish the peer review history of their article (what does this mean?). If published, this will include your full peer review and any attached files.

Reviewer #2: No

---

## [Author Response · Author response to Decision Letter 2]

3 Jun 2023

Please see our uploaded file displaying our responses to the reviewer.

---

## [Editor Report · Decision Letter 3]

5 Jun 2023

The epidemiology of HIV population viral load in twelve sub-Saharan African countries

PONE-D-22-26006R3

Dear Dr. Hladik,

We’re pleased to inform you that your manuscript has been judged scientifically suitable for publication and will be formally accepted for publication once it meets all outstanding technical requirements.

Kind regards,

José Antonio Ortega, Ph.D.

Academic Editor

PLOS ONE

Additional Editor Comments (optional):

It is felt that the revision has addressed the comments of the reviewer and that the manuscript fulfills PLOS ONE criteria for publication.
---

## [Editor Report · Acceptance letter]

9 Jun 2023

PONE-D-22-26006R3 

The epidemiology of HIV population viral load in twelve sub-Saharan African countries 

Dear Dr. Hladik:

I'm pleased to inform you that your manuscript has been deemed suitable for publication in PLOS ONE. Congratulations! Your manuscript is now with our production department. 

Kind regards, 

on behalf of

Dr. José Antonio Ortega 

Academic Editor

PLOS ONE